# PANDAA intentionally violates conventional qPCR design to enable durable, mismatch-agnostic detection of highly polymorphic pathogens

Iain J. MacLeod [1,2✉], Christopher F. Rowley[1,2,3] & M. Essex[1,2]

Sensitive and reproducible diagnostics are fundamental to containing the spread of existing and emerging pathogens. Despite the reliance of clinical virology on qPCR, technical challenges persist that compromise their reliability for sustainable epidemic containment as sequence instability in probe-binding regions produces false-negative results. We systematically violated canonical qPCR design principles to develop a Pan-Degenerate Amplification and Adaptation (PANDAA), a point mutation assay that mitigates the impact of sequence variation on probe-based qPCR performance. Using HIV-1 as a model system, we optimized and validated PANDAA to detect HIV drug resistance mutations (DRMs). Ultra-degenerate primers with 3′ termini overlapping the probe-binding site adapt the target through site-directed mutagenesis during qPCR to replace DRM-proximal sequence variation. PANDAA-quantified DRMs present at frequency ≥5% (2 h from nucleic acid to result) with a sensitivity and specificity of 96.9% and 97.5%, respectively. PANDAA is an innovative advancement with applicability to any pathogen where target-proximal genetic variability hinders diagnostic development.

[1] Department of Immunology and Infectious Diseases, Harvard TH Chan School of Public Health, Boston, MA, USA. [2] Botswana-Harvard AIDS Institute Partnership, Private Bag, Gaborone, Botswana. [3] Division of Infectious Diseases, Beth Israel Deaconess Medical Center, Boston, MA, USA. ✉email: iam391@mail.harvard.edu

Highly sensitive molecular diagnostics are fundamental to the control and clinical management of existing and emerging pathogens. As exemplified in the ongoing SARS-CoV-2 pandemic, in the absence of an efficacious treatment or vaccine, rapid identification of infected patients is the only control measure available to curb transmission and ensure timely containment of the infectious disease. The use of PCR and real-time PCR (qPCR)-based methodologies in clinical diagnostic virology demands high sensitivity and specificity for the selected viral nucleic acid target. Analytical specificity comprises the extent of inclusivity by which all viral phylogenetic variants are captured at the exclusion of its genetic near neighbors. This is facilitated by directing primer and probe design to evolutionarily conserved regions identified through multiple sequence alignments that portray geographical and temporal genomic variability. Subsequent primer and probe designs are selected primarily on two interdependent factors: the oligonucleotide melting temperature ($T_m$) – governed principally by oligo length and GC content – and its complementarity with the target nucleic acid.

Potentially uncharacterized genetic variation in the oligonucleotide-binding sites arising from ongoing viral evolution contributes to reduced assay efficiency or complete failure. The average sensitivity of published qPCR assays for RNA viruses, which accumulate considerable diversity over a short timeframe[1], has been predicted to be approximately 70%[2]. Sequence divergence within oligonucleotide-binding sites creates primer–template duplex instability and lowers $T_m$. This can be offset by increasing the primer length or GC content to confer a degree of mismatch tolerance or mitigated using degenerate nucleotides. These approaches must balance the amplification of heterogeneous quasi-species with non-specific amplification by off-target mispriming[3]. qPCR probe design is more averse to similar modifications because increasing probe $T_m$ through nucleotide degeneracy or sequence lengthening may lead to high $T_m$ variations that reduce specific target discrimination. Importantly, neither approach can account for future de novo allelic variants arising in the oligonucleotide-binding sites. Frequent in silico re-evaluation is therefore necessary to identify escape variants that necessitate assay primer and probe design, assay re-optimization, and clinical validation.

Attempts to circumvent these labor-intensive re-design processes include performing multiple qPCR assays in parallel (e.g., Lassa fever virus)[4] or sequentially (e.g., Crimean-Congo hemorrhagic fever virus)[5] to capture the majority of phylogenetic lineages and mitigate diagnostic errors on clinical interpretation. Yet dependency on multiple assays brings its own substantial time and economic burden that produces a bottleneck during an epidemic response to WHO high-priority pathogens[6]. These design complexities have been compounded during the unprecedented SARS-CoV-2 pandemic as rapid diagnostic development was paramount despite the extremely limited availability of SARS-CoV-2 genomic data, which in some instances used as few

as six sequences for primer and probe design[7]. SARS-CoV-2 evolution has now been shown to negatively affect qPCR diagnostic assay performance[8–10], and this is exemplified by the identification of a single novel polymorphism in the SARS-CoV-2 E Gene as associated with failure of the Roche cobas® SARS-CoV-2 E-gene qRT-PCR[11].

There is an urgent need for novel approaches that tolerate sequence diversity to safeguard assay inclusivity while maintaining exclusivity. Universal/pan-lineage diagnostics would increase global diagnostic harmonization and remove the reliance on lineage-specific assays confined to distinct geographies. To overcome the limitations of conventional qPCR, we systematically and intentionally violated canonical qPCR design principles that had remained unchallenged since their inception despite decades of reagent development[12–15]. We developed Pan-Degenerate Amplification and Adaptation (PANDAA), an innovative point mutation assay that addresses high genomic variability by normalizing probe-binding regions. PANDAA uniquely tolerates de novo sequence diversity, ensuring that diagnostic integrity is maintained throughout an epidemic or pandemic response.

As the prototypic fast-evolving RNA virus, HIV-1 represents one of the largest PCR diagnostic hurdles to surmount. As a model system, we optimized and validated PANDAA to detect single-nucleotide variations (SNVs) associated with HIV drug resistance (HIVDR). As HIVDR mutations (DRMs) occur at key genomic positions[16], we were constrained to predefined regions and could not target more conserved genomic regions. HIVDR emerges from the continuation of an antiretroviral treatment (ART) regimen in the absence of virological suppression. This limits the efficacy of current and future regimens by rendering one or more of the antiretrovirals (ARVs) – or even whole drug classes – ineffective[5]. The presence of a single DRM yields a high predictive value of reduced ART efficacy and treatment failure[16] and genotyping of six codons can detect major (non-)nucleoside reverse transcriptase inhibitors ([N]NRTI) DRMs in >98% of patients failing treatment[16]. Although resistance genotyping allows clinicians to classify virological failure as resistance- or adherence-mediated, and to select an alternative ART regimen conferring the highest likelihood of virological success, in low- and middle-income countries (LMICs), Sanger-based resistance genotyping is restricted until a patient fails two standardized regimens. Thus, a focused genotyping HIVDR diagnostic would address a profound gap in clinical care globally that is currently addressed ineffectively by more esoteric techniques.

## Results

**Principles of PANDAA amplification and adaptation.** We challenged the notion that extensive genetic heterogeneity of HIV-1 precludes the development of a universal qPCR diagnostic for resistance genotyping[17] by intentionally violating the core tenets of qPCR oligonucleotide design (Table 1)[18]. Using ultra-degenerate

**Table 1 PANDAA improves sensitivity to detect single nucleotide changes in HIV-1 compared to conventional qPCR.**

| Design condition | Primers | | Probes | |
| --- | --- | --- | --- | --- |
| | **Conventional qPCR** | **PANDAA** | **Conventional qPCR** | **PANDAA** |
| Length | 15–30 nt | 30–40 nt | 15–30 nt | 12–15 nt |
| Oligonucleotide $T_m$ | 50–60 °C | 65–75 °C | 68–70 °C | 58–60 °C |
| Primer–probe distance | 50 nt | Overlap | 50 nt | Overlap |
| 3′ terminus | ≤2 GC in the last 5 bp no mismatches | Mismatches encouraged | - | - |
| Nucleotide degeneracy | Avoid | Up to 40,000-fold* | Avoid | Permissible |
| GC content | 30–80% | 30–80% | 30–80% | 30–80% |

*must be empirically determined.

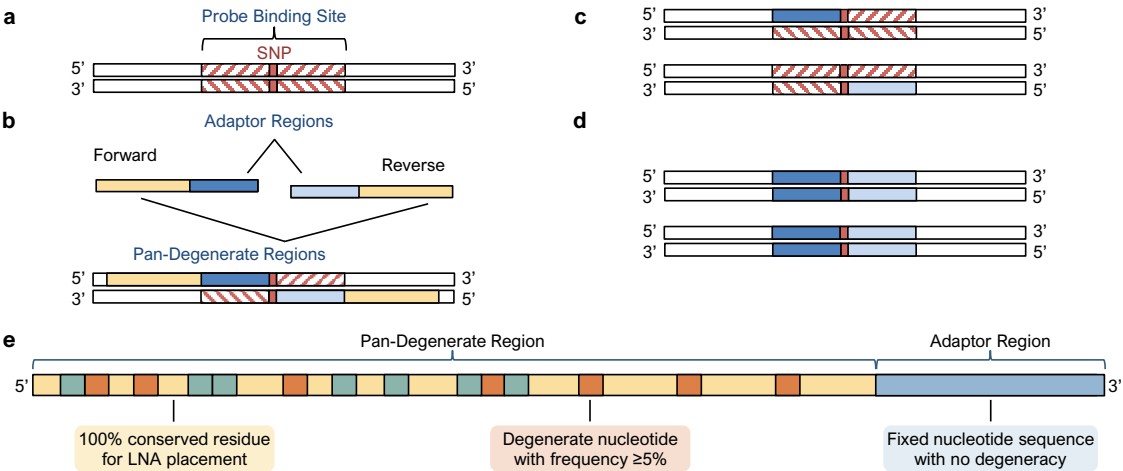

**Fig. 1 Overview of PANDAA method and primer design. a** Heterogeneous genomes contain single-nucleotide variants within the probe-binding site that prevent the use of probe-based qPCR for DRM discrimination. **b** PANDAA primers overlap with the probe-binding site, adapting the nucleotides proximal to the DRM of interest that would otherwise abrogate probe hybridization. **c, d** As qPCR proceeds, the newly generated amplicon will contain probe-binding sites flanking the DRM that are perfectly complementary to the probe. **e** PANDAA primers contain two key features: a 3′ ADR that is matched to the probe-binding site and a pan-degenerate (PDR) region that incorporates the nucleotide degeneracy observed in the primer-binding site of the target. The PDR accounts for the high degree of variability in primer-binding sites. LNA bases at 100% conserved positions in the PDR offset the thermodynamic instability of mismatches between the primer ADR and the template.

primers with 3′ termini overlapping the probe-binding site, the target nucleic acid is adapted through site-directed mutagenesis during the initial qPCR cycles to replace sequence variation within the probe-binding site flanking the primary drug resistance mutation (Fig. 1a–d). This generates an amplicon population with a homogenous probe-binding site whereby the only point of nucleotide variation is the DRM. PANDAA primers contain a 3′ adaptor region (ADR) matched to the probe-binding site and a pan-degenerate region (PDR) incorporating degenerate bases representative of nucleotide variability in the primer-binding site upstream of the ADR. PANDAA primers include locked nucleic acids (LNAs), which act as molecular anchors to increase primer affinity for their target and counter the thermodynamic instability of mismatches within the ADR (Fig. 1e).

**Determination of PANDAA oligonucleotide-binding site variability.** Conventional degenerate oligonucleotide design, using all available sequences, or an equivalent number of sequences from each subtype, to determine the consensus sequence, generally overestimates naturally occurring genomic variation as each variant nucleotide is incorporated as a discrete change. This generates oligonucleotides that do not occur naturally. PANDAA differs by considering the complete primer- or probe-binding site as a discrete genomic allele. Using codon 65 in HIV-1 reverse transcriptase (RT) as an example, the 95% consensus sequence for a 15-nt probe would generate 16 probe sequences (Supplementary Table 1). This is reduced to six sequences using our allele-based algorithm using HIV-1 subtype prevalence (Supplementary Table 2) to determine the probability of encountering a given subtype. By weighting primer- and probe-binding site allele frequency using subtype prevalence, the likelihood of encountering a matched target is increased. By contrast, an uncorrected approach that uses an equal number of sequences from each subtype to determine the most frequent primer-/probe-binding site, introduces bias from low-prevalence subtypes, particularly circulating and unique recombinants.

**PANDAA probe design validation.** We initially validated PANDAA to discriminate the wild-type amino acid lysine (K) of codon 103 in HIV-1 RT from the DRM arginine (N) arising from

an A → C transversion at the third nucleotide. Using approximately 95,000 unique patient-derived HIV-1 sequences, the probe sequence is constructed from the most prevalent probe-binding site allele (Supplementary Table 3). For PANDAA validation, we constructed DNA templates incorporating 19 probe-binding site alleles (Supplementary Table 3).

Canonical qPCR probes should have a $T_m$ 8–10 °C higher than the primer $T_m$, a constraint that purportedly ensures the probe out-competes primer hybridization during annealing[12,13]. To facilitate mismatch tolerance during adaptation, PANDAA primers have a $T_m$ 65–75 °C. Therefore, a PANDAA probe would have a $T_m > 75$ °C when applying canonical design rules, which would provide inadequate DRM nucleotide discrimination. We designed PANDAA probes with a $T_m$ near the 60 °C assay annealing temperature, favouring shorter probes that minimize the number of probe-binding site single-nucleotide variants to be adapted.

Using a DNA template with no probe-binding site mismatches (template 001), experimental validation indicated that K103N PANDAA probes can be reduced to as few as 13 nt by incorporating stabilizing nucleotide modifications (Fig. 2a–b). Although we disregarded the qPCR design principle prohibiting the overlap of primer- and probe-binding sites, PANDAA neither reduced qPCR amplification sensitivity nor negatively influenced specificity. Longer probes did not reduce amplification efficiency by out-competing the primer 3′ ADR for the overlapping probe-binding site on the same amplicon. Performance across 13–17 nt TaqMan-MGB probe lengths was equivalent (Fig. 2c) at a median Cq of 23.6 cycles at $10^4$ copies and 27.1 at $10^3$ copies. Equivalent yields of a 66-bp amplicon (Fig. 2d and Supplementary Fig. 1) confirmed that comparable qPCR performance was not an artefact of the higher $T_m$ and faster hybridization kinetics of longer probes potentially masking a reduction in amplicon yield and that complementarity between the probe and ADRs did not lead to the accumulation of non-specific products.

**PANDAA primer design validation.** To mitigate thermodynamic instability of 3′ primer ADR mismatches with the target, degenerate bases were incorporated along the PANDAA primer PDR at positions with nucleotide variability ≥5%. Using template

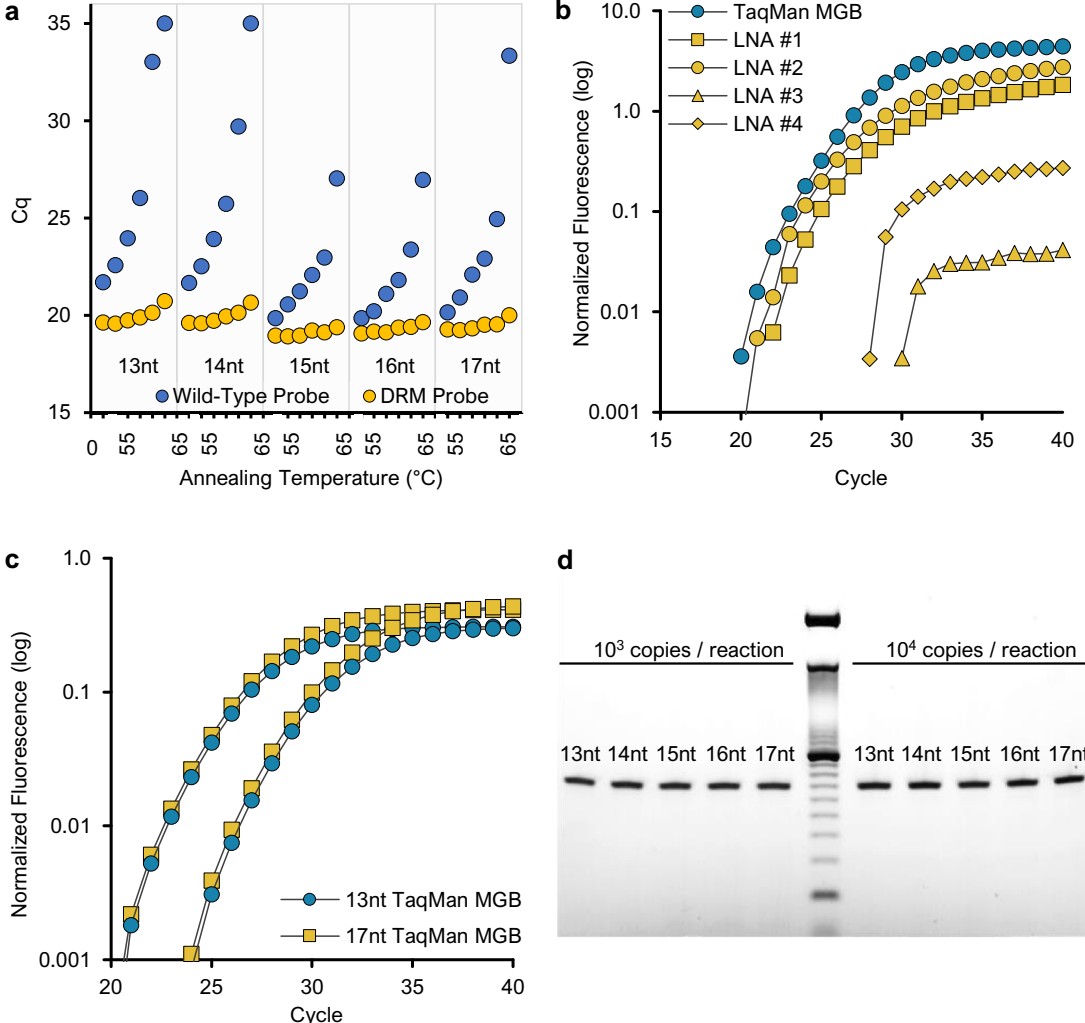

**Fig. 2 Optimization of PANDAA probe design. a** 13–17-nt TaqMan-MGB probes for 103 wild-type and DRM codons. Sensitivity and specificity were evaluated using a 55–65 °C annealing temperature gradient. Although predicted $T_m$ values ranged from 59 to 63 °C, there was little impact sensitivity with the AAC probes (yellow circles) for any probe length as the annealing temperature increased; however, non-specific hybridization of the AAA probe (blue circles) to the mismatched AAC template was reduced as annealing temperature increased. **b** The performance of four 5′ hydrolysis probes with various placement of LNA nucleotides (yellow) was compared to a TaqMan-MGB probe (blue) of the same length. LNAs were found to have comparable performance to MGB probes although LNA nucleotide placement also had to be determined empirically. **c** Increasing probe length did not reduce sensitivity, as 13-nt probes (blue circles) had similar Cq values to 17-nt probes (yellow) at $10^4$ and $10^3$ copies per reaction. **d** To confirm that amplification efficiency was not proportionally inhibited as the number of overlapping nucleotides increased, PANDAA reactions utilizing 13–17-nt probes were resolved on a 4% agarose gel, demonstrating comparable band intensities of a 66-bp amplicon across all probe lengths at both $10^3$ and $10^4$ DNA copies per reaction (middle lane: 10-bp DNA ladder). Furthermore, no non-specific products were evident.

001, degenerate PANDAA primers 2830 F and 2896 R improved sensitivity compared with non-degenerate consensus primers (Fig. 3a–b and Supplementary Table 4). Both primer sets contained the same 3′ ADR. We determined the tolerance for primer degeneracy up to 19,968-fold by incorporating degenerate bases in the PDRs to represent 95–99% of primer-binding site alleles (Supplementary Table 5). Using synthetic DNA with a single probe-binding site mismatch in both the forward and reverse ADRs (template 014), 2830F-96/97% forward with the 2896R-99% reverse primer increased sensitivity by approximately 22-fold (Fig. 3c and Supplementary Table 6). 2830F-99% exhibited no improvement in sensitivity compared with 2830F-95%, which may be due to a reduction in effective primer concentration relative to primer degeneracy: 2830F-99% (18,432-fold) compared with 2830F-96/97% (1,536-fold) (Supplementary Table 5). SYBR green experiments resolved single amplicon peaks suggesting that

the reduced amplification efficiency of 2830F-99% did not arise from reaction component sequestration by non-specific product formation (Fig. 3d and Supplementary Fig. 2). A similar pattern was observed when using the 001 template (Supplementary Table 6). Finally, increasing PDR degeneracy to incorporate co-expressed DRMs (i.e., those proximal to the primary DRM) as additional degenerate positions did not reduce performance (Fig. 3e).

Using two homogenous templates with different 2830 F and 2896 R primer-binding sites (Supplementary Fig. 3), single-clone sequencing ascertained that adaptation occurred across the PDR binding site at positions containing degenerate nucleotides, demonstrating a broad spectrum of degenerate PANDAA primer utilization (Fig. 3f and Supplementary Fig. 3). These sequences represent the predominate populations at the qPCR completion; therefore, those containing multiple substitutions may not

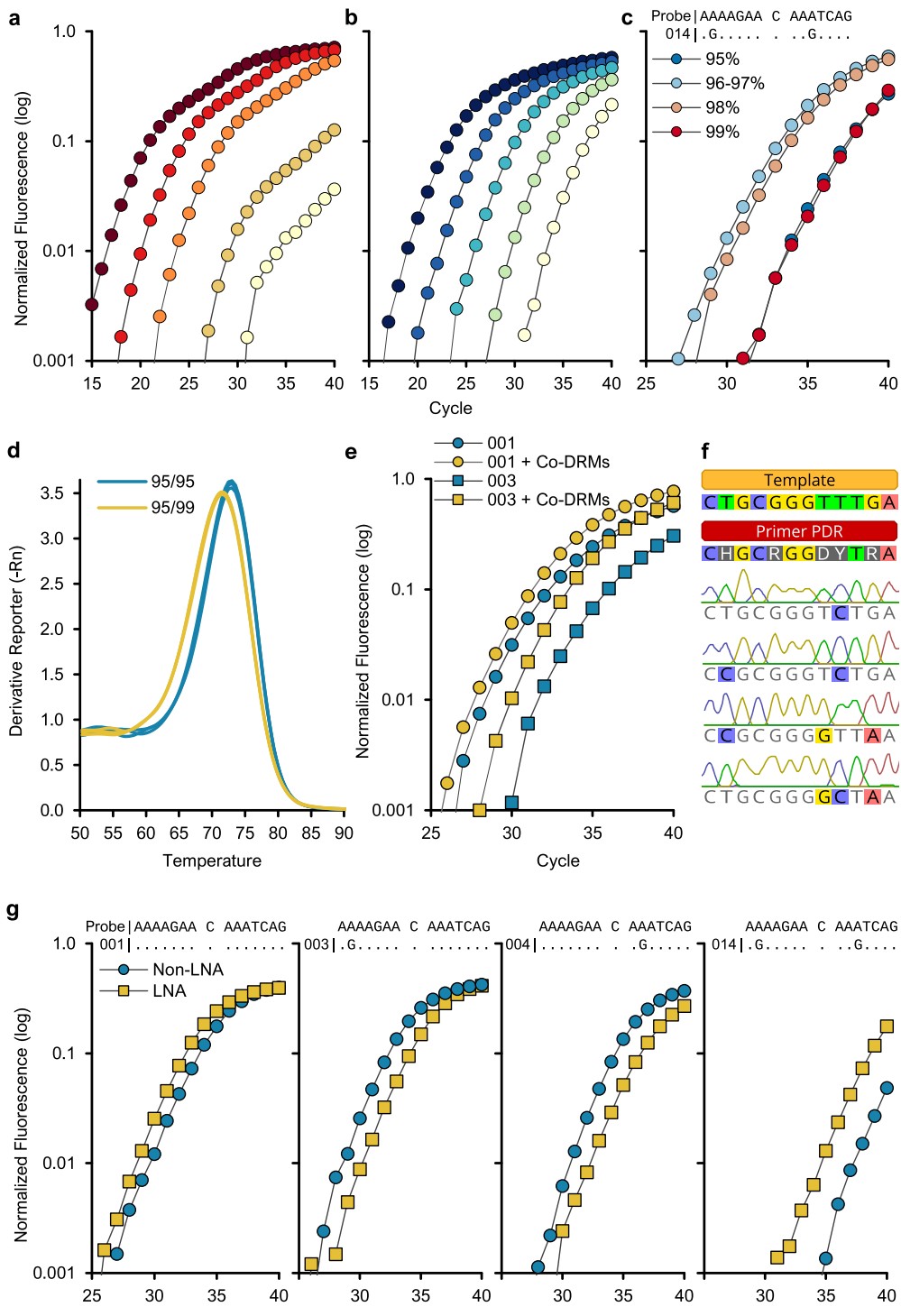

represent an adaptation to that sequence by a single degenerate primer. Rather, substitutions will have occurred in a stepwise manner, with one or two changes incorporated at a time. PDR adaptation increases the effective primer concentration with each cycle as progressively more primer variations in the degenerate pool can hybridize with newly adapted amplicon.

Lengthening primers to compensate for $T_m$ reductions arising from primer 3′ ADR-target mismatches would increase primer degeneracy further due to the high genomic heterogeneity of HIV-1. As an alternative strategy, the PDR incorporated LNA nucleotides at 100% conserved positions, which further enhanced

PANDAA sensitivity (Fig. 3g). Together, these iterative refinements of PANDAA primer design—the empirical determination of both optimal degeneracy and LNA placement at preferred positions—culminate in an assay that is highly tolerant of primer–template mismatches and eliminates probe-binding site sequence variation, which have constrained conventional qPCR design for decades.

**Resolution of probe-binding site mismatches.** Using PANDAA primers lacking the 3′ ADR, we compared conventional qPCR

**Fig. 3 Optimization of PANDAA primer design.** Consensus versus PANDAA primers. DNA template 001 was diluted from $10^6$ to 100 copies per reaction and was amplified using either the **a** majority consensus primers with no degenerate bases or **b** PANDAA primers. Both sets of primers contained the same 3′ ADR. A single mismatch was present in both the forward and reverse consensus primers (Supplementary Table 4). Using the K103N DRM-specific probe, both primer sets showed similar sensitivity down to $10^4$ copies; however, the lowest copy number quantified using consensus primers was $10^4$ copies compared to 100 copies for the PANDAA primers. Although <$10^4$ copies were detectable using consensus primers, the copies were not accurately quantifiable. PDR degeneracy, co-expressed DRMs, and melt curve: **c** Using $10^3$ copies of template 014, which contains a mismatch in each ADR, optimal PDR degeneracy was determined empirically. **d** The 2830F-99% PDR variant with either the 2896R-95% (blue) or 2896R-99% (yellow) variant in a SYBR green qPCR with the 014 template showed a similar relationship between PDR degeneracy and reaction efficiency to that with probe-based PANDAA. The lower amplicon $T_m$ and broader melt curve with the 2896R-99% primer reflected the wider range of predicted GC% content compared to 2896R-95% (25.8–58.1% versus 29.0–54.8%, respectively), and therefore the wider primer $Tm$ range (63.0–78.7 °C versus 64.2–77.3 °C, respectively). **e** Additional degeneracy was introduced at co-expressed DRMs included codons 100 and 101 in primer 2830 F and 106 in 2896 R. Performance was evaluated using template 001 (circles) or template 003 (squares). Primers with co-expressed DRMs (yellow) compared to the previous iteration of PANDAA primers (blue) were found to improve sensitivity. **f** Single-clone sequencing after PANDAA on a homogeneous DNA template demonstrated that adaptation was occurring within the PDR at positions containing degenerate bases. **g** Inclusion of LNA nucleotides at 100% conserved positions in the PDR increased sensitivity in templates containing probe-binding site mismatches in the forward ADR, reverse ADR, or both.

**Table 2 PANDAA improves sensitivity to detect single nucleotide changes in HIV-1 compared to conventional qPCR.**

| Allele | Probe-binding site | qPCR median Cq (IQR) | PANDAA median Cq (IQR) | ΔCq (PANDAA - qPCR) |
|---|---|---|---|---|
| Probe | AAAAAGAA C AAATCAGC | | | |
| 001 | ................ | 22.9 (22.6–23.0) | 22.6 (22.5–22.6) | −0.3 |
| 002 | ...........G..... | 29.5 (29.4–29.5) | 23.0 (23.0–23.0) | −6.5 |
| 003 | ..G.............. | 31.9 (31.9–32.0) | 26.2 (26.1–26.2) | −5.7 |
| 004 | .........G...... | n.d. | 22.7 (22.4–22.8) | – |
| 005 | ...C............ | 26.2 (26.1–26.2) | 23.4 (23.4–23.6) | −2.7 |
| 006 | .............A. | 24.1 (24.0–24.1) | 22.5 (22.5–22.5) | −1.6 |
| 007 | ....G........... | 25.1 (25.0 - 25.1) | 23.6 (23.5–23.7) | −1.5 |
| 008 | ......G......... | n.d. | 24.9 (24.8–25.0) | – |
| 009 | .G.............. | 23.9 (23.8–24.0) | 23.7 (23.5–23.9) | −0.2 |
| 010 | .............T.. | 25.4 (24.9–25.4) | 22.8 (22.8–22.9) | −2.6 |
| 011 | ..G..A.......... | n.d. | 32.6 (32.5–32.6) | – |
| 012 | .............C.. | 24.7 (24.3–24.8) | 22.9 (22.9–23.1) | −1.8 |
| 013 | .............G.. | 41.2 (40.3–41.4) | 22.6 (22.6–22.7) | −18.6 |
| 014 | ..G......G..... | n.d. | 23.1 (23.0–23.1) | – |
| 015 | .C.............. | 34.1 (34.1–34.1) | 24.6 (24.3–24.8) | −9.5 |
| 016 | ..C............ | 39.8 (39.6–40.1) | 25.8 (25.7–25.8) | −14.0 |
| 017 | ......G......A. | n.d. | 26.2 (26.2–26.3) | – |
| 018 | ...C.....G...... | n.d. | 23.2 (22.9–23.3) | – |
| 019 | ..G..........A. | n.d. | 33.1 (33.0–33.1) | – |

*n.d.* not detected.

with PANDAA for 19 probe-binding site variants (Supplementary Table 3). PANDAA increased the sensitivity by a median Cq of 2.7 cycles for all templates regardless of the position or number of mismatches (Table 2 and Fig. 4a–c). Where a single-nucleotide variant completely inhibited conventional qPCR, probe binding, and DRM detection were rescued by PANDAA to within a median Cq of 2.3 cycles from the perfectly matched template, 001 (Table 2). Differences between conventional qPCR and PANDAA were unlikely due to differences in amplification efficiency; both had similar median Cqs for template 001, which does not require adaptation. Thus, PANDAA can adapt to one or more positions with nucleotide variation in the K103 probe-binding site, independent of the mismatch position relative with to the DRM nucleotide, and the type of mismatch. Single-clone sequencing of PANDAA amplicons verified that adaptation occurred in the probe-binding site (Fig. 4d). Furthermore, PANDAA was successful in a one-step RT–qPCR using RNA, confirming that PANDAA does not impede cDNA synthesis (Supplementary Fig. 4).

PANDAA primer 2830 F contains two ADR mismatches at positions −3 and −6 in template 011 (Fig. 4e). We hypothesized that PANDAA performance would improve through sequential adaptation of each mismatch by including of low-concentration of 2830 F[−3A], which retains the −3G:A (template:primer) mismatch while adapting the −6 A:G, and 2830 F[−6G], which retains the −6A:G mismatch while adapting the −3G:A (Fig. 4e). This would generate a heterogeneous amplicon pool during the initial qPCR cycles whereby a proportion of amplicon would be adapted to match the probe-binding site only at the −3 position, and the remaining amplicons adapted only at the −6 position. The template pool would then contain amplicons with a single 2830 F PANDAA primer mismatch, allowing PANDAA to complete adaptation more efficiently (Fig. 4e). Relative to no sequential adaptation, sensitivity was improved by both the 2830 F[−3A] (1.8-fold) and 2830 F[−6G] primers (19.7-fold) (Fig. 4f) in a dose-dependent manner (Fig. 4g). The −3G:A mismatch was preferentially adapted given that 2830 F[−6G] sequential adaptation performed better than 2830 F[−3A]. Combining both sequential adaptation primers at an equimolar concentration was less effective (13.5-fold) compared with the −6G primer alone (Fig. 4f).

Sequential adaptation primer 2830 F[−6G] increased sensitivity by 4.9-fold for other probe-binding sites with the −6A:G mismatch, with a −1.5-fold reduction in amplification of non

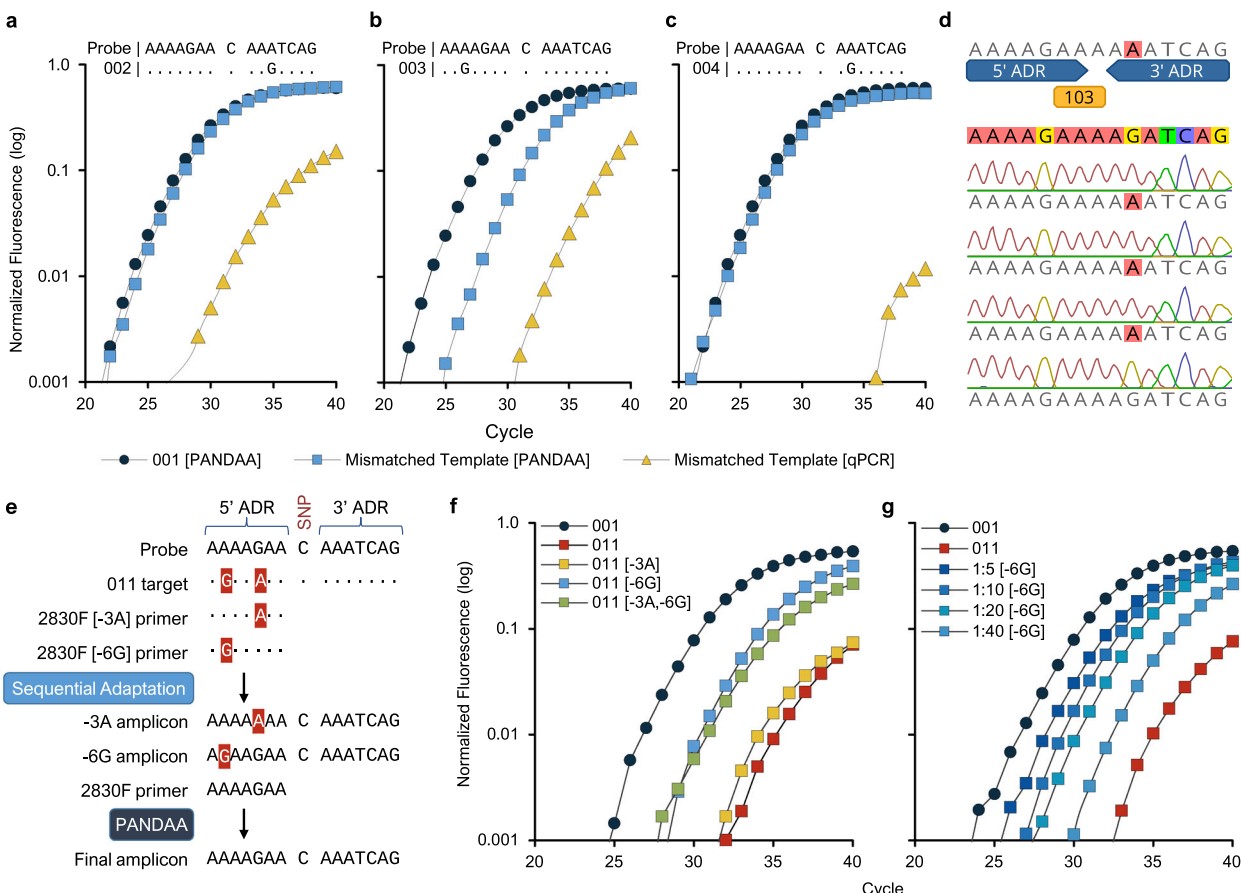

**Fig. 4 PANDAA adaptation of the probe-binding site rescues qPCR performance.** Conventional qPCR using PANDAA primers lacking the 3' ADR was compared to PANDAA for the four most common K103 probe-binding sites (001 to 004).Template 001 (blue circles), which does not contain probe-binding site sequence variation, was included as a reference in each experiment against which the target containing a single-nucleotide variation was compared when using PANDAA (blue squares) versus conventional qPCR (yellow triangles). **a** PANDAA restored detection of template 002 to the same level as a target with no probe-binding site sequence variation. **b** Detection of template 003 was increased almost 40-fold when using PANDAA compared to conventional qPCR. **c** Conventional qPCR did not detect template 004, whereas PANDAA restored detection close to that observed with template 001. **d** Single-clone sequencing of the PANDAA amplicon demonstrated adaptation of the probe-binding site when using template 004 as the target. Adaptation was seen in 42/44 clones (95.5%). **e** Sequential adaptation overcomes multiple ADR mismatches. Template 011 contains two mismatches in the 5' ADR, which must be adapted by the forward primer. By including a low concentration of 2830 F primers that contain only one of the two mismatches, adaptation can be performed in a stepwise manner such that the 2830 F PANDAA primer must only adapt one nucleotide variant rather than two. **f** Two primers, 2830 F [−3A] and 2830 F [−6G], each of which contains only a single mismatch, were added to separate PANDAA reactions at 10% of the 2830 F forward PANDAA primer concentration, and adaptation performance was evaluated with 103 copies per reaction template 001 or 011 DNA. 2830 F [−3A] resulted in a 0.9-cycle decrease compared to the 011 template with only the standard PANDAA primer. 2830 F [−6G] reduced the Cq by 4.8 cycles, whereas both sequential adaptation primers together led to a 4.1-cycle decrease. **g** A dose response was evident with 2830 F [−6G] sequential adaptation from 2.5 to 20% of the 2830 F forward PANDAA primer concentration. Results are the representative median of six replicates.

−6A:G templates (Supplementary Table 7). We surmised that this pro-amplification (pro-amp) effect arose from the partial decoupling of amplification from its dependence on adaptation during the initial qPCR cycles. By increasing the pool of un-adapted template, pro-amp offsets the amplification penalty linked to adaptation, such that a higher proportion of newly adapted amplicons can be generated within the initial qPCR cycles (Supplementary Fig. 5). We evaluated allele-specific pro-amp using low-concentration ADR-matched primers (eight 2830 F ADR variants and six 2896 R ADR variants), representing the 19 probe-binding site alleles. Sensitivity improved with individual pro-amp primers for each probe-binding site allele, with a median ΔCq of −1.0 cycles compared with without pro-amp (Supplementary Table 8). Pooling individual pro-amp primers negated this modest increase in sensitivity, with a median ΔCq of 0.0 cycles.

**Optimized PANDAA sensitivity, specificity, and selectivity.** We applied these refinements to PANDAA design for the K65R, Y181C, and M184VI DRMs to produce a highly specific, focused genotyping resistance assay for NNRTI-based ART regimens. We validated these PANDAA assays using two sets of five DNA templates incorporating probe-binding site alleles, covering ≥95% of patients. Integrated WT 001-005 contained the wild-type codon, and Integrated DRM 001-005 contained DRM-conferring nucleotide substitutions for K65R, K103N, V106M, Y181C, M184VI, and G190A (Supplementary Table 9). V106M and G190A were included in Integrated DRM templates as they overlap with the primer- and probe-binding sites for K103N and M184VI, respectively. PANDAA quantified both wild-type and the K103N DRM across a linear range to five copies ($r^2 = 0.998$) (Fig. 5a–c) with the other three DRMs performing similarly. PANDAA selectivity (the detectable DRM proportion on a wild-

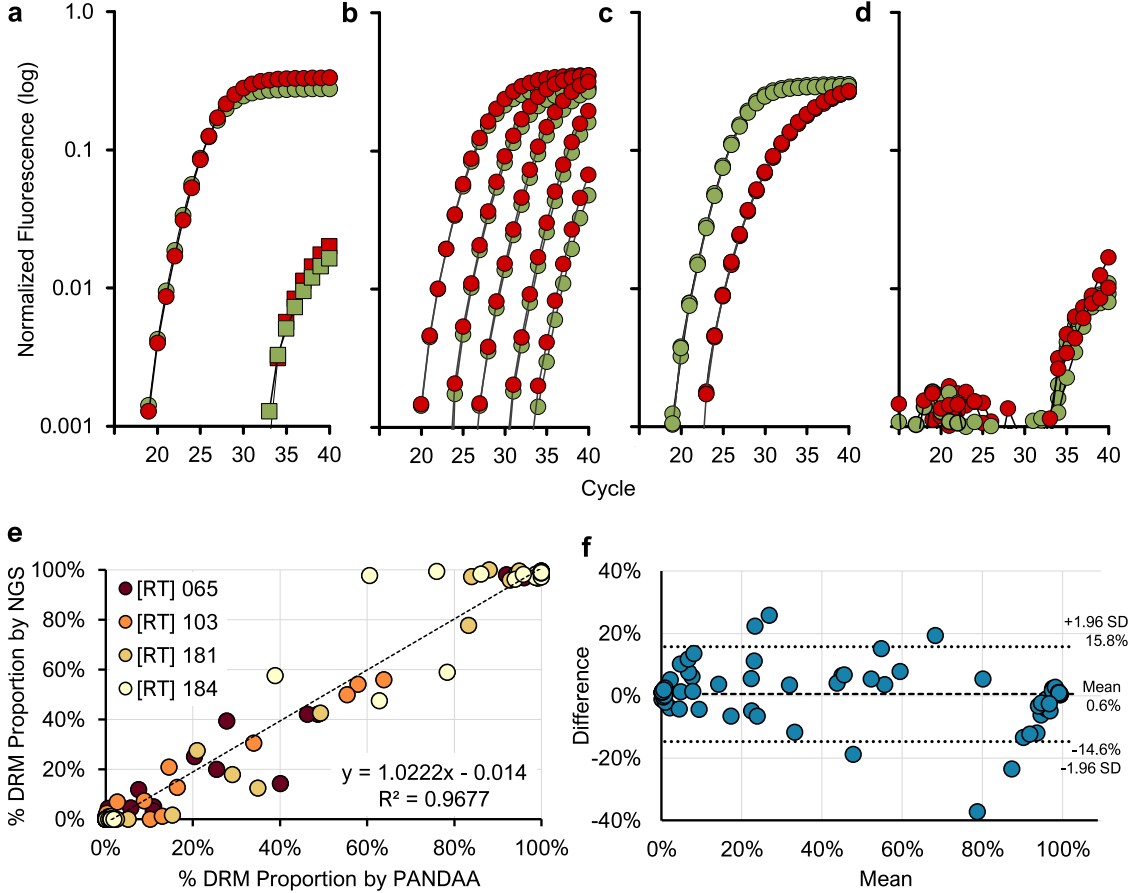

**Fig. 5 Validation of PANDAA performance.** PANDAA with differentially labeled TaqMan probes to discriminate wild-type DNA (VIC-labeled [green]) from the K103N DRM (FAM-labeled [red]). **a** Using either 100% wild-type DNA or 100% mutant 014 template DNA at $10^5$ copies per reaction, non-specific DRM signal (red squares) can clearly be differentiated from the specific wild-type signal (green circles) in wild-type only reactions. Similarly, the non-specific wild-type signal (green squares) are distinguishable from the specific K103 DRM signal (red circles). **b** PANDAA on 10-fold dilutions of a 1:1 mixture of $10^5$ to 10 total DNA copies; thus, wild-type (green circles) and mutant DNA (red circles) were present at 50% of those quantities. **c** Mixed populations of wild-type (green) to mutant DNA (red), representing 10% K103N DRM at $10^5$ total copies of DNA. **d** Negative control using human genomic DNA. Results are representative of a minimum of six replicates of each dilution series. The x-axis represents the number of qPCR cycles and the y-axis represents the log normalized fluorescence. **e** Correlation of PANDAA with NGS. Pearson's correlation coefficient showed a strong agreement between the K65, K103N, Y181C, and M184VI DRM proportions quantified by PANDAA and those quantified by NGS ($r = 0.9837$; 95% CI: 0.9759–0.9890; $P <$ 0.0001). **f** A Bland–Altman plot of the agreement between DRM quantification by PANDAA and NGS shows a mean bias of 0.6% (±7.8%) with 95% limits of agreement (dotted lines) ranging from −14.6% to 15.8%.

type background) was assessed using mixed ratios of Integrated WT and Integrated DRM 001-005 DNA templates down to a 1% DRM proportion. Extensive specificity evaluations using human genomic DNA indicated that all PANDAA assays maintained high specificity in the presence of highly complex, non-HIV nucleic acid (Fig. 5d).

**Clinical resistance genotyping by PANDAA compared with population sequencing and NGS.** We next evaluated PANDAA using 72 clinical samples from patients with virological failure on NNRTI-based ART. All samples were genotyped previously by population sequencing, and probe-binding site mismatches were present in 18–43% (Supplementary Table 10). Diluted PCR amplicons stored from population sequencing were focused genotyped by PANDAA for the K65R, K103N, Y181C, and M184VI DRMs. PANDAA had an excellent overall agreement with population sequencing at 97.6% concordance (Table 3 and Supplementary Table 11) and 100% concordance for Y181C and M184VI. Three samples that were genotyped as K65R by PAN-DAA yet wild-type by population sequencing had approximately

5–9% electrophoretic mixtures when evaluated using Geneious. For the four discordant K103N results, the proportion of DRMs as determined with PANDAA was 11–15%, close to the cut-off used for population sequencing. PANDAA had 96.9% sensitivity and 97.5% specificity in accurately classifying patients as first-line ART failures, defined as the presence of ≥1 six failure-defining DRMs (Supplementary Table 12). By drug class, PANDAA detected all patients with NRTI failure and 87.5% of those with NNRTI failure (Table 4 and Supplementary Table 12). We analyzed a subset of 25 samples using NGS to quantify DRM relative abundance and allow a comparison with the quantitative readout from PANDAA. Strong agreement was observed when PANDAA was compared with NGS for all four DRMs with Pearson's correlation coefficient ($r = 0.9837$; $P < 0.0001$) (Fig. 5e–f).

## Discussion
Sensitive and reproducible molecular diagnostics are a key control measure in containing the spread of existing and emerging pathogens. Despite the reliance of clinical virology on qPCR methodologies, technical challenges persist that compromise their

**Table 3 Agreement between PANDAA and population sequencing for four DRMs.**

| Codon | Wild-type | | DRM | | Agreement | Kappa (95% CI) |
|---|---|---|---|---|---|---|
| | Sanger | PANDAA | Sanger | PANDAA | | |
| 65 | 54 | 51 | 18 | 21 | 95.8% | 0.895 (0.778–1.00) |
| 103 | 61 | 59 | 11 | 13 | 94.4% | 0.813 (0.635–0.991) |
| 181 | 48 | 48 | 24 | 24 | 100% | 1.00 |
| 184 | 57 | 57 | 15 | 15 | 100% | 1.00 |
| All | 220 | 215 | 68 | 73 | 97.6% | 0.935 (0.887–0.983) |

**Table 4 Diagnostic sensitivity and specificity of PANDAA to determine first-line ART and drug class-specific failure.**

| | Sanger DRMs | Patient coverage | PANDAA DRMs | Sensitivity (95% CI) | Specificity (95% CI) |
|---|---|---|---|---|---|
| First-line failure | K65R, K103N, V106M, Y181C, M184VI, G190AS | 99.2% | K65R, K103N, Y181C, and M184VI | 96.9% (84.3–99.5%) | 97.5% (87.1–99.6%) |
| NRTI failure | K65R and M184VI | 98.7% | K65R and M184VI | 100.0% (85.7–100%) | 93.9% (83.5–97.9%) |
| NNRTI failure | K103N, V106AM, Y181C, and G190AS | 97.1% | K103N and Y181C | 87.5% (71.9–95.0%) | 100.0% (91.2–100%) |

reliable and sustainable epidemic containment. Sequence instability in probe-binding regions gives rise to false-negative results despite the generation of a specific amplicon. Here we describe the design, validation, and evaluation of PANDAA, which addresses these shortcomings and the technical limitations. PANDAA adapts the probe-binding site to mitigate the negative impact of sequence variability on qPCR performance, thus enabling sensitive and specific detection when conventional qPCR would have failed[19,20]. Using HIV-1 as a model system, PANDAA quantified drug resistance mutations regardless of the number or position of sequence variants. We demonstrated the robustness of PANDAA adaptation by showing that multiple points of nucleotide variation in the same ADR can be adapted sequentially by including a limiting concentration of single-mismatched primer.

By systematically violating codified design principles of qPCR, we demonstrated the flexibility of three canonical rules. First, that primer- and probe-binding site overlap and sequence complementarity do not impede amplification or generate spurious non-specific products. Rules forbidding this were inherited from qPCR design employing probes longer than those used here. By minimizing probe length, we reduced the number of probe-binding site sequence variants to be adapted. Theoretically, primer–probe hybridization can generate a non-specific amplicon incorporating the complete probe-binding site. With PANDAA, complementary exists only with the primer ADR of the opposite orientation, e.g., the first 7 nt of the sense-oriented probe is complementary to the antisense primer 3′ ADR. Unfavorable hybridization of so few nucleotides at the 60 °C annealing temperature, and the absence of the probe 3′ terminal hydroxyl group, reduce the likelihood of artificially generating the probe-binding site.

Second, the thermal instability of primer 3′ mismatches in the PANDAA ADR can be offset using LNAs to increase the $T_m$[21]. Any primer–template mismatch within the last four or five nucleotides of the 3′ terminus disrupt the DNA polymerase active site and are detrimental to primer extension[22,23], yet DNA polymerase variants, which have varying levels of 3′ mismatch extension efficiency[24,25], are overlooked in design guidelines. We leveraged the high rate of non-specific nucleotide extension from

3′ mismatched bases of *Taq*, which is a critical design consideration for PANDAA[24]. Additionally, with RNA templates, adaptation using the primer ADR can be delegated to the lower stringency cDNA synthesis step. Although amplification and adaptation efficiencies are interdependent, once adaptation has occurred, no primer–template mismatches are present in the newly synthesized amplicon, allowing subsequent amplification rounds to proceed with increasing efficiency.

Finally, PANDAA primers tolerate extreme degeneracy while ensuring low non-specific product formation. Increasing degeneracy is assumed to prematurely plateau the amplification by lowering the concentration of unique target-matched primers able to prime amplification, which will be consumed earlier in the reaction. With PANDAA, amplification was not limited to primers that are perfectly complementary to the target; ultra-degenerate primers with PDR-template mismatches participated in productive amplification, which was promoted by including LNAs at empirically determined conserved positions in the PDR. This facilitates participation by an increasing proportion of the degenerate primer pool in the reaction, further reducing the availability of degenerate primers to form non-specific products with the net effect of enhancing PANDAA reaction efficiency.

For patients with HIV infection, a uniform standard of care for those accessing treatment is unattainable using existing genotyping diagnostics as they cannot withstand the resource and technical constraints of clinical laboratories in LMICs. Without appropriate action, HIVDR will significantly undermine the global response to the HIV epidemic. Our work represents a major advancement in diagnostic development, and we aim to empower centralized laboratories with the ability to implement focused resistance genotyping as a reflexive diagnostic after a detectable viral load. PANDAA confers several advantages that support this goal. The geographic disparity related to the high failure rate of sequencing assays for non-B subtypes[26,27] is addressed by the subtype-independent universality of PANDAA. As our design algorithm incorporates either global or regional subtype prevalence data, an assay incorporating local HIV-1 sequence diversity in a specific geographical region can be readily designed, which may facilitate local research efforts in LMICs.

**Table 5 Strengthening of diagnostics for epidemic prevention and containment using PANDAA.**

| PANDAA design feature | Benefit |
|---|---|
| PANDAA can adapt de novo probe-binding site sequence variants. | An a priori understanding of all probe-binding site variations does not need to be known. Impact of position and nucleotide substitution type on sensitivity can be determined in advance. |
| PANDAA is a lineage- / strain-agnostic assay by virtue of the adaptation process. | Global diagnostic harmonization without the need for individual assays to address geographical genomic variability. |
| PANDAA's inherent tolerance of extreme primer degeneracy favors the development of multiplexed assays. | Detect and differentiate multiple subtypes in a single reaction e.g., differentiation of circulating influenza virus from pandemic strains[25,45]; reduced need to update diagnostics based on seasonal genetic drift. |

Once PANDAA is optimized for probe-binding site adaptation, there is substantial interchangeability between targeting a single DRM (e.g., M184V) or multiple DRMs at a codon (e.g., M184I/V) without the need for primer re-design or re-optimization. DRM-specific probes can be labeled with the same fluorophore if there is no clinical utility in differentiating between DRMs. This intrinsic flexibility removes redundant or superfluous detection reagents to maximize sample throughput and reduce costs. We showed that PANDAA quantifies DRMs present at ≥5% to return a focused genotyping result in approximately 90 min using RNA in a one-step RT–qPCR. With the excellent agreement between PANDAA and both population sequencing and NGS for four major RT DRMs, PANDAA has a diagnostic sensitivity and specificity of 96.9 and 97.5%, respectively, in patients with first-line ART failure using conventional population sequencing. The superior selectivity of PANDAA to detect low-frequency DRMs, below the 15–20% threshold of Sanger[28], is an additional strength to further improve patient outcomes[29,30]. This study does have several limitations. Although PANDAA was optimized using synthetic templates from multiple HIV-1 subtypes, only patient samples from subtype HIV-1C were available. A direct comparison using prospectively collected samples from independent cohorts in multiple geographical regions would evaluate the relative benefits and clinical utility of PANDAA compared with existing genotyping methods. This iteration of PANDAA was optimized for four DRMs; however, ongoing studies are expanding PANDAA to two additional first-line DRMs and for DRMs conferring resistance to second-line protease inhibitors. We are currently investigating multiplexed PANDAA for HIV-1 resistance genotyping.

More broadly, PANDAA can strengthen epidemic preparedness by insuring against the ongoing evolution of viral pathogens in many ways (Table 5). By validating PANDAA designs using putative probe-binding site sequence variants that are predicted to have the greatest impact on assay sensitivity, which we have shown is negligible (Table 2). This would significantly bolster clinical diagnostics against the risk of false negatives from uncharacterized de novo genetic variation in the oligonucleotide-binding sites. This has previously led to severe sensitivity loss in commercial influenza A assays[31] and has quickly rendered diagnostics all but obsolete, such as was seen with the 2009 CDC that saw mismatches arising in the primer- and probe-binding sites within three years[32]. Although there are limitations to the assumptions that we can make, we believe that the development of PANDAA as a multiplexed assay and its independent validation in resource-limited settings will establish it as a platform diagnostic technology for other highly polymorphic pathogens.

## Methods

**Study design**. The objective of this study was to develop a rapid genotyping assay, PANDAA, for HIV-1 drug resistance mutations using qPCR. This required an in-silico analysis of primer- and probe-binding site allele frequencies across all HIV-1 subtypes using a novel approach to weight allele frequency based on the global subtype distribution. DRM-discriminating TaqMan-MGB probes of various lengths were designed using the most common probe-binding site allele, and the optimal length was determined empirically using synthetic DNA and RNA templates representative of DRM-proximal sequence variation. Degeneracy was incorporated within the primer PDR to ensure ≥95% coverage of primer-binding site alleles, and optimal degeneracy was determined empirically. The inclusion of thermostabilizing nucleotide modifications was assessed to improve PANDAA sensitivity and specificity. Optimized PANDAA primer and probe designs were compared with conventional qPCR to quantify the improved sensitivity of PANDAA in the presence of 19 probe-binding sites containing mismatches at various positions relative to the DRM. Further evaluations of the limit of detection for PANDAA to quantify four DRMs were performed using synthetic DNA templates that represented ≥95% of probe-binding site alleles. Finally, PANDAA was compared with population and next-generation sequencing using deidentified PCR amplicons derived from previous genotyping workflows from patients failing a first-line NNRTI-based ART regimen.

**HIV-1 sequence alignment**. We searched the Los Alamos HIV public database (http://www.hiv.lanl.gov) for sequences within the genomic region 2550 → 3501 (HXB2 coordinates) from all subtypes – including recombinants – with a minimum fragment length of 500 nt. We selected a single sequence per patient, resulting in 93,611 sequences at the time of this study. Subtyping was determined with sequence-associated information from the Los Alamos database. Multiple sequence alignments were constructed using MAFFT 7.38[33] in Geneious 11.1 (http://www.geneious.com) and checked manually. HIV-1 RT DRMs within the alignment were determined using the Stanford University HIV Drug Resistance Database (http://www.hivdb.stanford.edu) and reverted to the wild-type codon sequence[17].

**Determination of primer- and probe-binding site allele frequencies**. Alignments were analyzed using a custom resequencing program written in Visual Basic (v7.1, Microsoft). The target region was extracted from each sequence and arrayed by subtype: A, 01_AE, 02_AG, B, C, D, F, and G. All other subtypes, including CRFs and URFs, were grouped as "Other". Sequences with deletions or ambiguous nucleotides were excluded. Unique target sequences within each subtype array were identified, and their prevalence was determined. Any unique sequence with a prevalence <0.5% was excluded as a potential sequencing error. The intra-subtype allele frequency ($f_{allele}$) is then adjusted based on the subtype prevalence ($p_{subtype}$): $f_{allele}$ x $p_{subtype}$. The final weighted prevalence of each target region allele is its cumulative frequency across all subtypes.

**Probe design**. For probes with an odd number of nucleotides and a single, centered DRM nucleotide, the upstream and downstream regions within the probe-binding site are of equal length: $\frac{n-1}{2}$ nucleotides where $n$ is the probe length. To ensure that the discriminating DRM nucleotide is biased toward the hydrolysis probe 3′ terminus, sense-oriented probes with an even number of nucleotides, have an upstream region $\frac{n}{2}$ nucleotides, and downstream region $\left(\frac{n}{2}\right) - 1$ nucleotide. For antisense probes, the region lengths are swapped. Probes to detect the DRM were labeled with a FAM fluorophore and those for wild type with VIC.

**DRM discrimination relative to probe length**. Discrimination of the K103N AAC DRM was evaluated using conventional qPCR with PANDAA primers lacking the 3′ ADR using $10^4$ copies per reaction of DNA template 001, which encodes the K103N DRM and does not contain additional probe-binding site sequence variation (Supplementary Table 3). DRM discrimination relative to probe length was determined empirically given the inaccuracy of TaqMan-MGB probe $T_m$ predictions. Results represent median of six replicates. PANDAA primers with complete ADRs overlapping the probe-binding site were used to evaluate the competition between the primers and probes using $10^4$ and $10^3$ copies per reaction of DNA template 001.

**PANDAA primer design**. A minimum of six PDRs of 30–40 nt were chosen for each forward and reverse primer-binding site. The 5′ terminal nucleotide would be placed at, or adjacent to, a conserved position at which an LNA nucleotide was incorporated. Two additional LNAs were placed downstream of 5′ terminus at 100% conserved positions based on previously reported design considerations[21,34,35]. The 95–99% consensus sequence was determined from the primer-binding site alleles with a cumulative frequency ≥95%. Primer ADR sequences were incorporated to represent the upstream and downstream regions of the optimal probe described above. Balanced ADRs are those for probe-binding sites of an odd-numbered length such that both PANDAA primer ADRs will be $\frac{n-1}{2}$ nucleotides. Final primer $T_m$ predictions were calculated using Oligo Analyzer Version 3.1[36], and a minimum of 36 pairwise primer combinations were empirically evaluated for optimal LNA placement and PDR degeneracy.

**PANDAA**. PANDAA was performed using an ABI 7900 (Applied Biosystems). Briefly, a 10-μL reaction contained 5 μL of reaction buffer (Kapa Probe Fast, Kapa Biosystems), and forward and reverse PANDAA primers, VIC-labeled wild-type, and DRM-specific FAM-labeled probes. PANDAA reactions were incubated at 95 °C for 3 min, followed by 10 three-step adaptation cycles of 95 °C for 3 s, 50 °C for 60 s, and 60 °C for 30 s; then 35 two-step amplification cycles of 95 °C for 3 s, and 60 °C for 90 s during which fluorescence data were captured. Reactions using RNA templates contained 15U MMLV reverse transcriptase (NEB), with an additional incubation step of 42 °C for 15 min. SYBR qPCR was performed under the same conditions in the absence of PANDAA probes using Kapa SYBR Fast (Kapa Biosystems) with a melt curve stage included in the qPCR cycling protocol. Technical replicate number depended on the final copies/reaction: ≥$10^4$ ($n = 4$); ≥$10^3$ ($n = 8$); and <$10^3$ ($n = 12$). Human genomic DNA at 0.05 ng/reaction (Promega) was included as the non-target nucleic acid negative control ($n = 8$ replicates). Amplicons were resolved on 4% agarose EX e-gels (Thermo Fisher) with a TrackIt™ 10 bp DNA ladder (Invitrogen).

Raw qPCR fluorescence data were exported from Applied Biosystems SDS software. Background correction was performed using LinRegPCR[37]. For each target codon, PANDAA reaction efficiency was determined from standard curves of 1:1 mix of wild-type:DRM template across a dynamic range and calculated as efficiency $(E) = 10^{-1/\text{slope}} - 1$. The quantification threshold ($N_q$) was set at 0.05, which intersected with the exponential phase of the amplification curve for all targets at all copy numbers. This was used to determine the quantification cycle ($C_q$) (the fractional number of cycles needed to reach $N_q$). $C_q$ values were corrected for differences in probe-binding efficiencies to avoid biasing DRM proportion quantification due to asymmetric probe hybridization kinetics. The complete methodology for PANDAA data analyses can be viewed in the Supplementary Methods.

**Linearity, sensitivity, specificity, and selectivity**. DNA concentration was determined with optical density, and copy number was calculated using the molecular weight of the nucleic acid before diluting to fixed copy numbers. Wild-type and DRM templates were mixed at a 1:1 ratio to a total of $10^6$ copies/ μL and serially diluted two-fold to 8 copies/μL to determine PANDAA linearity for each target as well as the limit of detection (LoD). Mixed ratios to provide a final DRM proportion of 25%, 10%, 5%, 2.5%, and 1% were prepared in the same manner. The estimated LoD was determined using the lowest copy number, whereby 95% of the replicates are positive and can be distinguished from the negative.

**Resistance genotyping of patient samples**. This study used de-identified PCR amplicon from the *Bomolemo* study, an observational cohort designed to demonstrate the tolerability and virological response to a fixed-dose efavirenz/ tenofovir/emtricitabine ART regimen[38]. This study was conducted in Gaborone, Botswana, between November 2008 and July 2011 by the Botswana-Harvard AIDS Institute and the Botswana Ministry of Health from whom Institutional Review Board approval was received. Viral RNA was isolated from patient plasma samples at virological failure and genotyped by population sequencing, as previously described[39]. Population sequencing chromatograms were analyzed using the automated resistance genotyping platform ReCall, with nucleotide mixtures called when the electropherogram peak was ≥10%. All patient-derived amplicon samples were diluted 1:1000 in dH₂O supplemented with carrier tRNA from which 2 μL was used in a PANDAA reaction, performed in triplicate by two operators.

NGS was performed using the MiSeq system (Illumina) with coverage to detect HIV-1 variants in 1% of the virus population. MiSeq libraries were prepared using the patient-derived amplicon with the MiSeq sequencing run performed at the Harvard Biopolymers Facility. Sequence quality was assessed using FastQC, and QTrim was used to remove Illumina adapter sequences, reads below 36 bases, and leading/trailing low quality or N bases. Paired-end reads were assembled into an HIV-1 Group-M consensus reference using Geneious Read Mapper software and non-synonymous detected using automated SNV calling. Results were verified using PASeq v1.4 (https://www.paseq.org)[40].

**Oligonucleotides**. LNA-modified 5′ hydrolysis probes were synthesized by IDT and TaqMan-MGB probes by Thermo Fisher. LNA-modified primers were synthesized from Exiqon, whereas unmodified primers from IDT.

**Synthetic DNA**. Synthetic double-stranded DNA was designed to evaluate the sensitivity and specificity of PANDAA. The 5′ region contains the T7 RNA polymerase promoter. Immediately downstream of the T7 promoter, and at the 3′ terminus, we included optimized primer-binding sites for SYBR green confirmation and normalization of copy number across different templates (Supplementary Fig. 6). Lyophilized DNA (geneStrings, Life Technologies) were resuspended in TE buffer to obtain a template master stock to 5 ng/μL, which was quantified by fluorometry to provide an accurate stock concentration (Qubit dsDNA HS Assay Kit, Thermo Fisher). Templates were subsequently diluted in dH₂O supplemented with carrier tRNA from *Saccharomyces cerevisiae* at 0.05 μg/μL (Sigma Aldrich) to provide a dilution series from $10^6$ copies/μL to 5 copies/μL.

**Synthetic RNA**. In vitro transcription of single-stranded RNA used 25 ng synthetic DNA with the HiScribe T7 High Yield RNA Synthesis Kit (NEB). DNA template was removed from the RNA prep using RQ1 RNase-Free DNase and subsequently purified using RNeasy MiniElute Cleanup Kit (Qiagen; Hilden, Germany) with additional on-column DNA digestion. RNA was quantified by fluorometry to provide an accurate stock concentration (Qubit RNA HS Assay Kit, Thermo Fisher) and subsequently diluted in dH₂O supplemented with carrier tRNA to $10^6$ copies/μL. Serial dilutions were performed in the same manner as those for synthetic DNA templates.

**Single amplicon cloning and sequencing**. qPCR reactions were ExoSAP treated and purified (Wizard SV Gel and PCR Clean-Up System, Promega) before being ligated into the pMini vector (NEB PCR Cloning Kit, NEB). The ligation reaction was transformed into 10-beta Competent *E. coli* (NEB) and plated onto agar containing 100 μg/mL ampicillin. After overnight incubation at 37 °C, inserts were screened using colony PCR using the Cloning Analysis Forward and Reverse primers (NEB) to identify a minimum of 45 single amplicon clones, which were then sequenced using the same screening primers (Genewiz). By sequencing 45 clones, we had 95% confidence that we would detect sequence variants present in the amplicon population with a frequency of ≥10%: with $n$ single amplicons sequenced, the probability ($P$) of missing a variant after screening $n$ genomes is calculated as $f = 1 - (1 - P_s^{\frac{1}{n}})$ when the variant comprises a fraction $f$ (or less) of the virus population[41].

**PANDAA data analyses**. Normalizes probe-binding efficiencies to avoid bias due to asymmetric probe hybridization kinetics. As the efficiency of PANDAA primers to amplify a target region is independent of whether the codon of interest encodes a wild-type amino acid or DRM then any differences in qPCR efficiency determined to exist between wild-type and DRM detection must arise due to differences in probe-binding efficiencies arising from small $T_m$ variances due sequence differences at the target SNV. The efficiency correction factor ($E_{correction}$) is determined with Eq. 1[42,43]. The Cq of the DRM probe ($Cq_{(DRM)}$) is adjusted ($Cq_{(DRM)corrected}$) (Eq. 2) to that which would have been obtained had both probes had the same hybridization efficiencies.

Although the wild-type and DRM amplification curves become parallel after efficiency correction, there will still exist difference in target Cq despite target nucleic acid being present in equal proportions for both probes. This second adjustment factor arises due to differences in probe fluorophore characteristics (i.e., background fluorescence, signal-to-noise ratio)[43,44]. As target efficiencies have already been corrected, $Cq_{shift}$ can be readily determined (Eq. 3). Provided $Cq_{shift}$ is constant across a dynamic range of input copy numbers at an equal ratio of wild-type and DRM templates, then the final, adjusted DRM Cq is determined with Eq. 4[43,44]. This allows the proportion of DRM-harboring virus to be calculated using the common efficiency (Eq. 5).

$$E_{correction} = \frac{\log(E_{DRM})}{\log(E_{WT})} \tag{1}$$

$$Cq_{(DRM)corrected} = Cq_{(DRM)} \times E_{correction} \tag{2}$$

$$Cq_{shift} = Cq_{(DRM)corrected} - Cq_{(WT)} \tag{3}$$

$$Cq_{(DRM)shift} = Cq_{(DRM)corrected} - Med\left(Cq_{shift}\right) \tag{4}$$

$$Ratio = E_{WT}^{-\left(Cq_{(DRM)corrected} - Cq_{(WT)}\right)} \tag{5}$$

**Statistics and reproducibility**. Technical replicate number depended on the final copies/reaction: ≥$10^4$ ($n = 4$); ≥$10^3$ ($n = 8$); and <$10^3$ ($n = 12$). The agreement between genotyping methods was determined with Pearson's correlation coefficient. Kappa is a measure of the degree of non-random agreement between

observers or measurements of the same categorical variable. The agreement was considered as good at kappa 0.60–0.80 and very good at kappa >0.80.

**Reporting summary**. Further information on research design is available in the Nature Research Reporting Summary linked to this article.

## Data availability

The authors declare that the main data supporting the findings of this study are available within the article, and data sets for the main figures are in the Supplementary Data file. Due to the proprietary nature of the PANDAA technology, primer and probe sequences cannot be freely shared.

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

## Acknowledgements

C.F.R. was supported by NIH/NIAID Division of Microbiology and Infectious Diseases award R01 AI089350.

## Author contributions

I.J.M. and C.F.R. developed PANDAA and designed the performance experiments. I.J.M. was primarily responsible for data acquisition. C.F.R. fabricated the molecular clones. M.E. provided supervisory support. I.J.M., C.F.R., and M.E. discussed and interpreted the results and wrote and edited the manuscript.

## Competing interests

The authors declare the following competing interests: I.J.M. was an employee of the Harvard T.H. Chan School of Public Health at the time this research was performed and

is currently a co-founder, shareholder, and employee of Aldatu Biosciences, Inc, a diagnostics company that commercializes the PANDAA technology. Aldatu Biosciences, Inc had no role in the conceptualization, study design, data collection and analysis, and decision to publish or preparation of the manuscript. Patents relevant to this work include the following: US 10100349 B2 and EP 3052656 B1 ("Methods of determining polymorphisms") on which I.J.M., C.F.R., and M.E. are the inventors and are assigned to the President and Fellows of Harvard College, Cambridge, MA.
