## [Peer Review File · Communications Biology]

REVIEWERS' COMMENTS:

Reviewer #1 (Remarks to the Author):

The manuscript describes a highly innovative and novel approach to differentiating mutations from wild-type in the context of high background genetic variation. In this instance it was applied to HIV drug resistance mutations.

Although primer-determined template adaptation is a normal part of PCR when dealing with diverse templates, PANDAA appears unique in the intentional sequential adaptation to maintain efficiency and the adaptation of the probe binding region, to allow allelic discrimination – differentiation of particular wild type and mutant alleles despite a high background genetic variation in the primer and probe binding regions.

This manuscript overall contextualizes the problem adequately, provides a well-written account of the design, results and experiments; it uses figures and graphics appropriately to illustrate the principles. The conclusions are justified by the results provided.

I therefore regard this as a valuable scientific contribution that would be interesting to the journal readership.

Please address the following minor aspects.

Introduction:

1. The following statement in the text is misleading: “average sensitivity of published qPCR assays for RNA viruses ... as low as 26% for HIV-1 assays” this quotes an article from 2008 which reference one HIV-1 assay only and is not consistent with current evidence. Commercial assays have implemented innovative approaches to improve sensitivity; e.g the use of dual targets, primer degeneracy, design adaptation of probes. Indirect evidence that current commercial HIV viral load assays are sensitive is clear from very small fraction of HIV-1 seropositive patients, who are not on treatment who have undetectable viral loads, and amongst these the majority have evidence of natural viral load control (e.g. elite controllers, as evident from their disease course). Please correct and update this statement.

2. SARS-CoV-2, has been used as an example for dealing with viral diversity. Please clarify this, as this virus has a relatively slow evolution rate conserved (as the RNA polymerase has proofreading ability). (e.g. Oscar A MacLean, Richard J Orton, Joshua B Singer, David L Robertson, No evidence for distinct types in the evolution of SARS-CoV-2, *Virus Evolution*, Volume 6, Issue 1, January 2020, veaa034, <https://doi-org.ez.sun.ac.za/10.1093/ve/veaa034>]

Results:

3. “We determined the tolerance for primer degeneracy up to 19, 968-fold” Is “determine” the correct word; or should this mean “improved” ?

Reviewer #2 (Remarks to the Author):

This manuscript describes PANDAA (Pan-Degenerate Amplification and Adaptation), a point mutation that addresses high genomic variability by normalizing probe-binding regions with high sensitivity and specificity. The manuscript is very well written, and the work is original, underscoring a novel and innovative method that overcomes the limitations of conventional qPCR using HIV-1 as a

model system and tolerates de novo sequence diversity, important to rapid epidemic/pandemic intervention and response.

I only have 6 minor suggestions that I will leave up to the editors and authors to determine if they should be addressed and some suggestions to improve the manuscript:

Query 1: Abstract, page 1. When the author says “PANDAA-quantified DRMs present at $\geq 5\%$ ”, the sentence looks incomplete. The sentence could be edited, such as “PANDAA-quantified DRMs present at $\geq 5\%$ of frequency”.

Query 2: Abstract, page 1. The author talks about the time of the assay “2 h from nucleic acid to result” but through the results and discussion, this subject is no longer mentioned. Please consider discussing through this topic comparing the complexity and methodology benefits between qPCR and PANDAA.

Query 3: Results, page 2. To my knowledge “The Principles of PANDAA amplification and adaptation” should be in the topic Methods. Please consider moving this paragraph.

Query 4: Results, page 2. In the “The Principles of PANDAA amplification and adaptation”, the Table 2, shouldnt be Table 1?

Query 5: Material and Methods, page 12. “Optimized concentrations of forward and Reverse PANDAA primers”. For those who want to reproduce PANDAA experiment, it is important to inform which concentrations were tested.

Query 6: Material and Methods, page 13. “which were then sequenced using the same screening primers (Genewiz)”. Missing the reference for the sequencing reaction or the description of the sequencing.

Response to Reviewers' Comments

Reviewer #1

1. The following statement in the text is misleading: “average sensitivity of published qPCR assays for RNA viruses ... as low as 26% for HIV-1 assays” this quotes an article from 2008 which reference one HIV-1 assay only and is not consistent with current evidence. Commercial assays has implemented innovative approaches to improve sensitivity; e.g the use of dual targets, primer degeneracy, design adaptation of probes. Indirect evidence that current commercial HIV viral load assays are sensitive is clear from very small fraction of HIV-1 seropositive patients, who are not on treatment who have undetectable viral loads, and amongst these the majority have evidence of natural viral load control (e.g. elite controllers, as evident from their disease course). Please correct and update this statement.
 - I am unsure as to what the reviewer is referring to. The reviewer states that “current commercial HIV viral load assays are sensitive” and this is evidenced by the fact that there are HIV-1 patients with “undetectable viral loads”. By definition, if their viral load is undetectable then they cannot be used as a study population for determining the sensitivity of a viral load assay. Nevertheless, the reviewer makes my point for me – HIV assays have had to be continually improved due to increasing sequence diversity, which would not be an issue with PANDAA. Furthermore, although only a single HIV-1 assay was evaluated in that publication, the authors looked at 59 assays across a wide array of virus families.
 - Given the improvement in the performance of HIV-1 viral load assays, I have removed “as low as 26% for HIV-1 assays” from the text.
2. SARS-CoV-2, has been used an example for dealing with viral diversity. Please clarify this, as this virus has a relatively slow evolution rate conserved (as the RNA polymerase has proofreading ability). (e.g. Oscar A MacLean, Richard J Orton, Joshua B Singer, David L Robertson, No evidence for distinct types in the evolution of SARS-CoV-2, Virus Evolution, Volume 6, Issue 1, January 2020, veaa034, <https://doi-org.ez.sun.ac.za/10.1093/ve/veaa034>)
 - We have updated the introduction to include some references showing that SARS-CoV-2 evolution is already negatively affecting the performance of qPCR diagnostic assays.
3. “We determined the tolerance for primer degeneracy up to 19, 968-fold” Is “determine” the correct word; or should this mean “improved” ?
 - The word “determined” is correct.

Reviewer #2

Query 1: Abstract, page 1. When the author says “PANDAA-quantified DRMs present at $\geq 5\%$ ”, the sentence looks incomplete. The sentence could be edited, such as “PANDAA-quantified DRMs present at $\geq 5\%$ of frequency”.

- We have altered the abstract text to clarify that “PANDAA-quantified DRMs present at frequency $\geq 5\%$ ”.

Query 2: Abstract, page 1. The author talks about the time of the assay “2 h from nucleic acid to result” but through the results and discussion, this subject is no longer mentioned. Please consider discussing through this topic comparing the complexity and methodology benefits between qPCR and PANDAA.

- The use of RNA as a template is mentioned in the Results section under the heading “Resolution of probe-binding site mismatches” and the corresponding data are referenced in the text as Supplementary Figure 4.

Query 3: Results, page 2. To my knowledge “The Principles of PANDAA amplification and adaptation” should be in the topic Methods. Please consider moving this paragraph.

- The description of the PANDAA method is the central premise of the manuscript and as such it could result in a great deal of confusion for the reader should the principles section be moved to the Methods, which would disrupt the flow of the results.

Query 4: Results, page 2. In the “The Principles of PANDAA amplification and adaptation”, the Table 2, shouldnt be Table 1?

- We have confirmed that the sequential numbering of the tables is correct.

Query 5: Material and Methods, page 12. “Optimized concentrations of forward and Reverse PANDAA primers”. For those who want to reproduce PANDAA experiment, it is important to inform which concentrations were tested.

- We have removed the phrase “optimized concentrations” from the text. As the sequence designs are proprietary, the concentration information would be moot without the accompanying sequences. Nevertheless, the primers and probes in their optimized formats will be available from Harvard University and / or Aldatu Biosciences for those wishing to reproduce the experiments.

Query 6: Material and Methods, page 13. “which were then sequenced using the same screening primers (Genewiz)”. Missing the reference for the sequencing reaction or the description of the sequencing.

- We reference the sequencing primers used and their commercial source (“Cloning Analysis Forward and Reverse primers (NEB)”). Sequencing was outsourced to Genewiz as described. Given the ubiquity of Sanger sequencing in molecular biology, we don’t believe that an in-depth description of sequencing is necessary here and its absence does not detract from the quality of the manuscript.